# Increased Levels of cAMP by the Calcium-Dependent Activation of Soluble Adenylyl Cyclase in *Parkin*-Mutant Fibroblasts

**DOI:** 10.3390/cells8030250

**Published:** 2019-03-15

**Authors:** Paola Tanzarella, Anna Ferretta, Simona Nicol Barile, Mariella Ancona, Domenico De Rasmo, Anna Signorile, Sergio Papa, Nazzareno Capitanio, Consiglia Pacelli, Tiziana Cocco

**Affiliations:** 1Department of Basic Medical Sciences, Neurosciences and Sensory Organs, University of Bari “Aldo Moro”, 70124 Bari, Italy; tanzarellapaola87@gmail.com (P.T.); anna.ferretta@uniba.it (A.F.); mariellaa83@gmail.com (M.A.); anna.signorile@uniba.it (A.S.); sergio.papa@uniba.it (S.P.); 2Department of Biosciences, Biotechnologies and Biopharmaceutics, University of “Bari Aldo Moro”, 70124 Bari, Italy; simona.barile@uniba.it; 3Institute of Biomembranes, Bioenergetics and Molecular Biotechnologies, Italian National Research Council (CNR), 70126 Bari, Italy; d.derasmo@ibiom.cnr.it; 4Stazione Zoologica Anton Dohrn, 80121 Napoli, Italy; 5Department of Clinical and Experimental Medicine, University of Foggia, 71122 Foggia, Italy; nazzareno.capitanio@unifg.it

**Keywords:** parkin, mitochondria, cAMP, calcium

## Abstract

Almost half of autosomal recessive early-onset parkinsonism has been associated with mutations in *PARK2*, coding for parkin, which plays an important role in mitochondria function and calcium homeostasis. Cyclic adenosine monophosphate (cAMP) is a major second messenger regulating mitochondrial metabolism, and it is strictly interlocked with calcium homeostasis. *Parkin*-mutant (Pt) fibroblasts, exhibiting defective mitochondrial respiratory/OxPhos activity, showed a significant higher value of basal intracellular level of cAMP, as compared with normal fibroblasts (CTRL). Specific pharmacological inhibition/activation of members of the adenylyl cyclase- and of the phosphodiesterase-families, respectively, as well as quantitative reverse transcription polymerase chain reaction (RT-qPCR) analysis, indicate that the higher level of cAMP observed in Pt fibroblasts can contribute to a higher level of activity/expression by soluble adenylyl cyclase (sAC) and to low activity/expression of the phosphodiesterase isoform 4 (PDE4). As Ca^2+^ regulates sAC, we performed quantitative calcium-fluorimetric analysis, showing a higher level of Ca^2+^ in the both cytosol and mitochondria of Pt fibroblasts as compared with CTRL. Most notably, inhibition of the mitochondrial Ca^2+^ uniporter decreased, specifically the cAMP level in PD fibroblasts. All together, these findings support the occurrence of an altered mitochondrial Ca^2+^-mediated cAMP homeostasis in fibroblasts with the *parkin* mutation.

## 1. Introduction

Parkinson’s disease (PD) is one of the most common neurodegenerative diseases, characterized by the selective death of dopaminergic neurons in the *substantia nigra*. Although most cases of PD are sporadic, several genes involved in Mendelian forms of PD have been identified. Specifically, almost half of all the early-onset familial PD cases have been associated with mutations in *PARK2*, which codes for parkin, a multifunctional E3 ubiquitin ligase [1,2], which mediates ubiquitination of several target proteins [3]. Studies on in vitro and in vivo parkin-null models strongly suggest a role of parkin in the preservation of mitochondrial function. Parkin knockout mice exhibit mitochondrial dysfunction and oxidative damage [4,5]. Furthermore Drosophila parkin null mutants display mitochondrial dysfunction and apoptotic muscle degeneration [6,7]. Functional assays in leukocytes [8], as well as the fibroblasts of patients with parkin mutations, consistently show mitochondrial impairment [9,10,11]. Parkin has emerged as an important factor in mitochondrial quality control mechanisms [12,13] and in calcium (Ca^2+^) homeostasis [14,15,16,17]. In previous studies, we showed that primary fibroblasts of two different *parkin*-mutant patients, affected by early-onset autosomal recessive PD, displayed severe ultrastructural abnormalities, (mainly in the mitochondria) [10], altered expression levels of several proteins involved in cytoskeleton structure dynamics, Ca^2+^ homeostasis, oxidative stress response protein, and RNA processing [18,19] as well as an altered lipidomic profile [20]. In addition, increased reactive oxygen species (ROS) production, impaired energy metabolism, and increased cyclic adenosine monophosphate (cAMP) level were found in *parkin*-mutant fibroblasts [10].

cAMP is one of the major second messengers that regulates mitochondrial metabolism [21,22,23,24,25,26,27], and it is strictly intertwined with Ca^2+^ homeostasis [28]. cAMP is produced both by the hormonally responsive G protein-regulated transmembrane adenylyl cyclases (tmACs) [29] and by soluble adenylyl cyclase (sAC) [22,30,31]; the latter is found in discrete compartments such as the nucleus, mitochondria, and centrioles [30,32,33,34]. Degradation of cAMP is accomplished by phosphodiesterases (PDEs) encompassing 11 distinct PDE gene families, which are grouped into three categories based on their substrate specificity [35,36,37]. Ca^2+^ is known to regulate sAC activity and that of specific tmACs [38,39,40] and its dysregulation impairs mitochondrial function, and this can lead to cell death [41].

In this study we show, in *parkin*-mutant (Pt) fibroblasts, a higher basal level of cAMP resulting from a greater contribution by sAC and a low expression of the isoform PDE4. Moreover, cytosolic and mitochondrial Ca^2+^ measurements in Pt fibroblasts show a high basal Ca^2+^ level in the cytosolic and, in particular, in the mitochondrial compartments resulting in increased mitochondrial Ca^2+^-mediated cAMP level.

## 2. Materials and Methods

### 2.1. Skin Fibroblast and Culture Conditions

Primary skin fibroblasts from three patients P1, P2, and P3 (or Pt), affected by early-onset PD, with different parkin compound heterozygous mutations (P1 and P2 with del exon2–3/del exon3 [10], and P3 with del exon7–9/Glu409X [11,19]), and from one healthy subject, (P1 and P2′s mother), were obtained by explants from skin punch biopsy. Adult normal human dermal fibroblasts, purchased from Lonza Walkersville Inc., (Walkersville, MD, USA) were utilized as an unrelated control. The study was approved by the local Ethics Committee at the University of Bari Medical School, and the patients gave informed consent in accordance with the declaration of Helsinki.

Cells were grown in high-glucose Dulbecco’s modified Eagle’s medium (DMEM) supplemented with 10% (*v*/*v*) fetal bovine serum (FBS), 1% (*v*/*v*) ʟ-glutamine, 1% (*v*/*v*) penicillin/streptomycin, at 37 °C in a humidified atmosphere of 5% CO_2_. All experiments were performed on cells with similar passage numbers, ranging from 5 to 14, to avoid an artefact, due to senescence, known to occur at passage numbers greater than 30. In the passage range used, the fibroblasts were β-Gal-negative.

### 2.2. Cyclic Adenosine Monophosphate Assay

The cAMP levels were measured using the direct cAMP ELISA Kit (Enzo Life Sciences, New York, NY, USA). The cells were treated in the presence of dimethyl sulfoxide (DMSO) (0.02%), used as control vehicle, or 10 µM Rolipram, 100 µM 3-Isobutyl-1-methylxanthine (IBMX), 10 µM forskolin, 1 µM isoproterenol, 100 µM SQ22536, 25 µM of KH7 or 1 µM carbonic anhydrase inhibitor (CAI) for 30 min at 37 °C where indicated in the figures. For cAMP assays, the culture medium was removed and 1 mL of 0.1 M HCl was added to the cell layer, followed by 10 min incubation at 37 °C. The lysed cells were scraped and transferred into Eppendorf tubes. The samples were centrifuged at 1300× *g* for 10 min at 4 °C, and the supernatants were used for cAMP measurements according to the manufacturer’s instruction. The measurements were performed on a Victor 2030 multilabel reader (PerkinElmer, Waltham, MA, USA)). The cAMP values were normalized to the protein concentration, and expressed as pmol/mg protein.

### 2.3. Laser Scanning Confocal Microscopy (LSCM) Imaging of Cytosolic and Mitochondrial Ca^2+^

CTRL and Pt primary fibroblasts were cultured onto 24 × 24 mm glass slides in six-well plates and incubated for 30 min at 37 °C, with the following probes: 2 µM Fluo-4 AM (Life Technologies, Carlsbad, CA, USA) for cytosolic Ca^2+^ and 4 µM X-Rhod-1 AM (Invitrogen, Molecular probes^TM^, Carlsbad, CA, USA) for mitochondrial Ca^2+^ level measurements. Stained cells were washed with Phosphate Buffered Saline (PBS), fixed in PBS containing 4% paraformaldehyde for 10 min at room temperature, and cell nuclei were stained with 1 μg/mL 4′,6-Diamidino-2-phenyindole, dilactate (DAPI) Sigma-Aldrich, St. Louis, MO, USA) for 10 min at room temperature. Stained cells were examined with a Leica TCS SP8 microscope (images collected using a 40× and 60× in oil immersion objective) coupled to the Laser Scanning Confocal Microscopy (LSCM) system. Acquisition, storage, and data analysis were performed using Leica software LAS AF 3 (https://www.leica-microsystems.com/products/microscope-software/).

### 2.4. Quantitative Fluorimetric Measurement of Cytosolic and Mitochondrial Ca^2+^ Levels

Cytosolic and mitochondrial Ca^2+^ concentration was measured by using, respectively, the fluorescent indicator Fluo-4 AM and X-Rhod-1AM (Invitrogen, Carlsbad, CA, USA). CTRL and Pt primary fibroblasts were grown in a T25 Flask. Cells at 80% confluence were incubated with a fluorescent probe for 30 min at 37 °C. Cell monolayers collected by trypsinization and centrifugation were resuspended in a buffer containing 10 mM HEPES and 6 mM d-Glucose (pH 7.4) at an approximate concentration of 1 × 10^5^ cells in 1 mL. Fluorescence intensity was measured at 25 °C in a spectrofluorometer (Jasco FP6200 Mary’s Court Easton, MD, USA), equipped with a stirrer and temperature control, by the subsequent addition of 5 mM CaCl_2_, 0,1% Triton X-100 (for cytosolic Ca^2+^ levels), 0.1% Na-Cholate (for mitochondrial Ca^2+^ levels) and 40 mM EGTA. The excitation/emission wavelengths were 495 nm/506 nm for Fluo-4 AM and 580 nm/602 nm for X-Rhod-1 AM. The cytosolic and mitochondrial Ca^2+^ levels were evaluated by using an apparent Kd (443 nM for Fluo-4AM and 700 nM for X-Rhod-1AM) according to the equation described by Grynkiewicz et al. [42]. Where indicated, incubation with 1 µM Thapsigargin, 10 µM Dantrolene, and 5 µM Ruthenium Red (RR) was performed for 30 min at 37 °C.

### 2.5. Real-Time PCR

The purification of total RNA from fibroblasts was carried out by using RNeasy Mini Kit (Qiagen, Venlo, The Netherlands), according to the manufacturer’s protocol. One microgram of total RNA was then reverse-transcribed to generate cDNA for PCR by using the iScript cDNA Synthesis kit (Bio-Rad, Hercules, CA, USA). Semi-quantitative determination of *ADCY* and *PDE* messenger RNA (mRNA) levels were performed by real-time qRT-PCR, using SYBR Green (Bio-Rad). Reactions were performed in duplicate for each sample in three independent experiments. Multiple reactions were performed in a volume of 25 μL containing 12.5 μL of 2 × PCR master mix, 0.2 μM of specific primers, and 200 ng of cDNA template. Amplifications were performed in the BioRad iCycler iQ Real-Time PCR Detection System (BIO-Rad, Hercules, CA, USA), using the following cycle program: denaturation step at 95 °C for 10 min followed by 40 cycles of denaturation at 95 °C for 15 s, annealing at 60 °C for 1 min, and extension at 72 °C for 30 s. The relative mRNA expression levels were calculated by using the comparative CT method (ΔΔCT) [43]. Quantitative normalization for each sample was performed by using glyceraldehyde-3-phosphate dehydrogenase (GAPDH) as an internal control. Validated primers for semi—qRT-PCR are provided in Appendix A.

### 2.6. Western Blot Analysis

Whole cell extracts (30 μg) were separated on a 13% Sodyum-Dodecyl-Sulphate Polyacrilamide Gel Electrophoresis (SDS-PAGE) according to [44], and transferred onto a nitrocellulose membrane. Western blot analysis was performed by using specified primary antibodies against cyclic AMP-responsive element binding protein (CREB) and phosphorylated-CREB (P-CREB) (1:1000; Santa Cruz Biotechnology, Dallas, TX, USA), according to the manufacturer’s suggested concentrations. Protein loading was assessed by reprobing the blots with GAPDH (1:3000; Santa Cruz Biotechnology). After incubation with the corresponding horseradish peroxidase-conjugated secondary antibody (1:3000; Cell Signaling Technology, Danvers, MA, USA) signals were developed using an enhanced chemiluminescence kit (ClarityTM Western ECL Substrate, Bio-Rad), acquired by ChemiDoc Imaging System XRS (BioRad), and analyzed for densitometry with the Image J Lab software 1.8.0_112 (https://imagej.nih.gov/ij/index.html).

### 2.7. Protein Measurement

Total protein concentration was determined by the Bio Rad Bradford protein assay, using bovine serum albumin as the standard.

### 2.8. Statistical Analysis

Data are shown as mean ± SEM. The significance of any differences throughout the data sets presented (i.e., treated samples vs. controls) was determined by one-way Analysis of Variance (ANOVA) with the Bonferroni post-hoc test and with Student’s t-test. The threshold for statistical significance (*p*-value) was set to 0.05.

## 3. Results

### 3.1. The Basal cAMP Content in Parkin-Mutant Fibroblasts Is Higher than in Control Fibroblasts

The second messenger cAMP is involved in many important cellular signaling pathways, and several of these have been demonstrated to be defective in *parkin*-mutant fibroblasts [10]. We previously reported that fibroblasts harboring a heterozygous PARK2 mutation (del exon2-3/del exon3) displayed a higher cAMP basal level, as compared to control cells associated with hyperphosphorylation of CREB [10]. Here, we confirmed these observations, and we extended the analysis to a second control sample and a third patient harboring a different mutation in the *parkin* gene (del exon7-9/Glu409X). Figure 1A shows a significant higher level of cAMP in the fibroblasts of each patient (P1 30 ± 6%, P2 56 ± 19%, P3 37 ± 7%,) as compared with the CTRL fibroblasts (shown as pooled values of two different control subjects). Consistently, the phosphorylated form of CREB, which is a downstream target of the cAMP/Protein Kinase A (PKA) pathway, was found to be increased compared with CTRL fibroblasts (Figure 1B). These results, confirming the previous data, suggest that mutations in *parkin* affect the cAMP metabolism, irrespective of the genetic background. To gain deeper insights into cAMP metabolism, we performed further experiments only on P3 fibroblasts, (later referred as Pt fibroblasts) as cellular model of parkin null-cells.

In mammals, cAMP is generated from ATP by adenylyl cyclases (ACs), and degraded by specific phosphodiesterases (PDEs). The treatment of fibroblasts with 10 µM Rolipram, a selective inhibitor of PDE4, the mostly expressed isoform in fibroblasts [45,46,47,48] led to a significant increase of the cAMP level (40 ± 6%) in CTRL samples (Figure 2A) without significant effects in Pt cells, thus suggesting a likely defective PDE4 activity. Conversely, the treatment of cells with IBMX, a non-selective PDE inhibitor, resulted in a significant increase of the cAMP content in both CTRL (29 ± 9%) and Pt (29 ± 7%) fibroblasts (Figure 2B). Together, these observations suggest defective PDE4 activity in Pt cells, which deserves further investigations.

Intracellular cAMP is synthesized either by the hormonally responsive, G protein-regulated transmembrane adenylyl cyclases (tmACs) [29], or by soluble adenylyl cyclase (sAC) [22,30], found in distinct compartments such as nucleus, mitochondria and centrioles [30,32,33], and that are sensitive to Ca^2+^ and bicarbonate cellular levels [31,32,49]. On this basis, we investigated the activity of tmACs, treating cells with forskolin, a plant diterpene which binds to the catalytic unit of tmACs, triggering a strong enzymatic response, and with isoproterenol, a beta receptor agonist that binds specific receptors at the plasma membrane level, with both not affecting sAC activity. As expected, FSK-treatment led to a significant increase in cAMP levels, in both CTRL (346 ± 47%) and in Pt cells (214 ± 36%) (Figure 2C). Conversely, treatment with isoproterenol, resulted in a significant increase in cAMP levels only in CTRL cells (556 ± 60%) (Figure 2D). To further dissect the specific contributions of tmACs and sAC, we treated cells with SQ22536 [50] and KH7 [51,52], inhibitors of tmACs and sAC, respectively. SQ22536-treatment reduced the overall cellular cAMP content, both in CTRL and Pt fibroblasts, by approximately 40% (Figure 3A), whilst KH7-treatment caused a much greater significant decrease of the cAMP level in Pt (47 ± 3%) than in CTRL (28 ± 2%) cells (Figure 3B). Notably, in the presence of KH7, the cAMP level in Pt, and CTRL cells showed comparable values (inset Figure 3B), whereas, in the presence of SQ22386, the cAMP level in Pt was much higher than CTRL cells, thereby suggesting a greater contribution from sAC (inset Figure 3A) to the cAMP content in Pt fibroblasts.

Since sAC is stimulated by bicarbonate [31] we also treated cells with acetazolamide (CAI) [53,54,55] a selective carbonic anhydrase inhibitor. As for KH7, CAI decreased the cAMP level significantly (Figure 3C) to a larger extent in Pt (61 ± 2%) than in CTRL (48 ± 2%). In this condition, cAMP level in Pt and CTRL cells showed comparable values (inset Figure 3C), suggesting, once again, a greater contribution from sAC in Pt fibroblasts.

### 3.2. The Enhanced Content of cAMP in Parkin-Mutant Fibroblasts is Linked to the Deregulated Expression of PDE4A and sAC

In order to assess whether the differences described in the cAMP basal level between CTRL and Pt fibroblasts could be due to differential gene expression of the main players in cAMP metabolism, i.e., adenylyl cyclase and phosphodiesterases, we analyzed their transcription levels. RT-qPCR analysis of mRNA transcript levels revealed that the PDE4A mRNA level was significantly lower in Pt fibroblasts, as compared with CTRL fibroblasts (Figure 4), whilst the level of ADCY10, which codes for the sAC, was 4-fold higher in Pt fibroblasts, as compared with CTRL cells (Figure 4). The analysis of the mRNA level of the genes coding for ADCY3 and ADCY6, widely expressed in various human tissues, showed no differences for the first one, and a significant decrease for ADCY6 (35 ± 5%) (Figure 4). These data confirm a defect of PDE4 in the Pt fibroblasts, and supports the hypothesis of a major contribution of sAC to the observed higher cAMP levels therein.

### 3.3. Cytoplasmic and Mitochondrial Ca^2+^ Contents are Higher in Parkin-Mutant Fibroblasts as Compared with Control Fibroblasts

Mammalian sAC is stimulated by Ca^2+^ in a calmodulin-independent way, by lowering the K_m_ for Mg^2+^ ATP [56]. In order to assess if the higher cAMP levels in the Pt fibroblasts was contributed to by the Ca^2+^-dependent activation of sAC, we measured the cytosolic and mitochondrial Ca^2+^ steady-state levels. The fluorescence intensity analysis of the confocal microscopy images in the presence of specific cytosolic (Fluo-4 AM) (Figure 5A) and mitochondrial (X-Rhod-1 AM) (Figure 5B) Ca^2+^ probes revealed significantly higher levels of both cytosolic and mitochondrial Ca^2+^ in Pt cells, as compared with CTRL fibroblasts (Figure 5A,B). Quantitative fluorimetric measurements confirmed higher cytosolic (25 ± 3%) (Figure 5C) and mitochondrial (+27 ± 3%) (Figure 5D) steady-state Ca^2+^ levels in Pt fibroblasts, compared with CTRL cells.

Afterward, we investigated whether the increased mitochondrial basal Ca^2+^ level measured in the Pt cells could result from altered Ca^2+^ fluxes from intracellular sources. Both the mitochondria and the endoplasmic reticulum (ER) are the major regulators of Ca^2+^ homeostasis, therefore controlling Ca^2+^-dependent cellular signaling. Treatment with RR, inhibitor of the mitochondrial Ca^2+^ uniporter (MCU), resulted, as expected, in an enhanced cytosolic (Figure 6A), and a reduced mitochondrial (Figure 6B), steady-state Ca^2+^ level, both in the CTRL and Pt cells, thus indicating an efficient mitochondrial Ca^2+^ uptake by this uniporter. Subsequently, to analyze the involvement of endoplasmic reticulum (ER) in Ca^2+^ homeostasis, and the mitochondrial and cytosolic Ca^2+^ levels were measured in the presence of thapsigargin (TG), a specific irreversible inhibitor of ER Ca^2+^-ATPase (SERCA) [57], which mediates Ca^2+^ uptake by the ER, and dantrolene, a specific inhibitor of ryanodine receptor (RyR), the major ER Ca^2+^ release channel [58]. As shown in Fig. 6C, TG treatment induced a significantly high increase of the cytosolic Ca^2+^ levels, both in CTRL (184 ± 3%) and Pt (158 ± 9%) fibroblasts, and a slight but significant increase of the mitochondrial Ca^2+^ levels in the CTRL cells (Figure 6D), in contrast to the small but significant decrease of Ca^2+^ levels observed in Pt fibroblast mitochondria after TG treatment. Bravo et al. have shown in TG-treated HeLa cells the redistribution of ER and mitochondria near the nucleus, and also an increased amount of reticular and mitochondrial connection points [59]. Unexpectedly, dantrolene treatment induced a modest though significant increase of the cytosolic Ca^2+^ levels, both in CTRL (20 ± 3%) and Pt (17 ± 3%) fibroblasts (Figure 6E), and a more consistent increase of the mitochondrial Ca^2+^ level, only in CTRL cells (Figure 6F).

These apparently contradictory results might be reconciled with the notion that: (i) the mitochondrial Ca^2+^ entry is largely contributed to by the ER at the ER–mitochondria contact sites, where the local concentration of Ca^2+^ is exceptionally higher than that which is reachable in the cytoplasm, and (ii) the major ER Ca^2+^ channel at the inter-organelle contact sites is contributed to by systems that are different from the RyR (e.g., the IP3 receptor) [60].

### 3.4. Inhibition of Ca^2+^ Uptake in Mitochondria Specifically Decreases cAMP Content in Parkin-Mutant Fibroblasts

To test the possible link between mitochondrial Ca^2+^ and the high level of cAMP, CTRL and Pt fibroblasts were treated with RR or Ru360, a more selective inhibitor of MCU, and the cellular cAMP content was assayed. The results show, for both the MCU inhibitor treatments, that while the CTRL cells showed a significant increase in cAMP content, in Pt fibroblasts, this was significantly reduced (Figure 7). These observations are consistent with our previous conclusion, according to which: (a) in CTRL cells’ cAMP levels are mainly contributed to by the activity of the cytosolic tmAC, which was stimulated by the rise in cytosolic Ca^2+^, following inhibition of MCU; (b) in *parkin*-mutant fibroblasts, the mitochondrial cAMP level was higher, and the overall cellular cAMP level was is affected more significantly by the partial inactivation of sAC, following the inhibition of mitochondrial Ca^2+^ uptake.

### 3.5. High Mitochondrial Ca^2+^ Level Driven by Dantrolene Treatment in CTRL Fibroblasts Induces an Increase in cAMP Content

As a proof of principle, we took advantage of the observed increase of mitochondrial Ca^2+^ induced by dantrolene in CTRL fibroblasts to verify, if independently from the parkin-null context, this was sufficient to cause an increased level of cAMP production. As shown in Figure 8A, dantrolene treatment of CTRL fibroblasts led to a significant cAMP level increase (31 ± 5%). Furthermore, the treatment of CTRL cells with dantrolene plus SQ22536, the inhibitor of tmAC, induced a significant increase of cAMP, compared to treatment with SQ22536 alone (Figure 8A). Notably, the extra cAMP content induced by dantrolene in the presence of SQ22536 was comparable with that observed in the absence of the tmAC inhibitor. thus leading to the conclusion that it was largely, if not exclusively, attributable to Ca^2+^-mediated activation of the mitochondrial sAC. This condition would mimic what constitutively observed in Pt fibroblasts, supporting the hypothesis that the observed higher cAMP level in the Pt fibroblasts is indeed the result of Ca^2+^-mediated activation of sAC.

In order to assess whether the high calcium level measured in dantrolene-treated CTRL might led to differential ADCY10 and PDE4A gene expression, as observed in Pt cells, we measured the mRNA levels. In this setting, the ADCY10 mRNA levels were significantly higher (Figure 8B), as well as the expression of PDE4A, possibly to counteract the observed cAMP increase, as one would expect in healthy cells.

## 4. Discussion

In mammalian cells, the cAMP produced in the cytosol by tmAC, acts as a second messenger (cAMP cascade), following binding to specific receptors of the extracellular signal molecules, like hormones and neurotransmitters [29]. On the other hand, sAC, which is widely expressed in different subcellular compartments, is activated by bicarbonate [22,55,61] and by Ca^2+^ [38].

In the present work, we confirmed the presence of high cAMP level in fibroblasts harboring a heterozygous PARK2 mutation (del exon2-3/del exon3) [10] and extended the analysis to a third patient harboring a different mutation in *parkin* gene (del exon7-9/Glu409X) [11]. The study was addressed to define factors such as adenylyl cyclase, phosphodiesterase, and cytosolic and mitochondrial Ca^2+^, which are causally involved in the observed high level of the basal cAMP in *parkin*-mutant fibroblasts (Pt cells), taken as a cellular model of parkin null-cells.

The direct activation of tmAC by FSK raised cyclic AMP levels, as expected, both in CTRL, and even if to a lesser extent, in Pt cells, whereas isoproterenol, a β-adrenergic receptor agonist coupled to stimulatory Gs proteins, which determine the activation of the transmembrane adenylate cyclase, resulted in a significant increase of cAMP levels, only in CTRL cells. The lack of effect by isoprotenerol in Pt cells, not due to an alteration of tmAC intrinsic activity, as shown by the responsiveness to FSK stimulation, deserves further investigation.

Alongside with the higher cAMP content in Pt cells, we showed that a specific lack of response occurred toward Rolipram, a specific PDE4 inhibitor, but not to IBMX, a PDEs pan-inhibitor. On the contrary, the treatment of CTRL fibroblasts showed a significant increase in the cAMP level, since the degradation path was inhibited not only by the PDE4 inhibitor, but also by IBMX treatment. As IBMX treatment resulted in similar level of cAMP increase in both the CTRL and Pt cells, we could deduce that similar PDE activity is present in both cell types, except for the PDE4A isoform, whose expression is reduced in Pt cells. This is in line with recent work showing that the loss of PDE4 expression in the striato-thalamo-cortical circuit, which is associated with deficits in spatial working memory, in PD patients [62].

In the present work, the results of treatment with KH7, a specific inhibitor of sAC, which has been shown to be able to suppress the mitochondrial cAMP content almost completely in isolated mitochondria [24], revealed a higher contribution of sAC to the basal cAMP level in Pt cells. This hypothesis is supported by the comparable effect that was obtained in the presence of acetazolamide, a CAI inhibitor. Of note, inside the mitochondrial matrix, sAC activity began in response to both metabolically generated CO_2_ [22], as well as Ca^2+^ entry [40], highlighting a potential crosstalk and synergy between these two common second messengers in regulating energy metabolism [63,64]. Notably, the expression level of ADCY10 was markedly upregulated in Pt fibroblasts, further supporting a large contribution of sAC to the observed elevated cAMP content therein.

The high basal levels of cAMP in the *parkin*-mutant fibroblasts were likely be a result of high levels of Ca^2+^ in the cytosol, which, buffered by the mitochondria, would in turn lead to the activation of mitochondrial sAC. Although the exact mechanism of action is yet to be established, studies on human SH-SY5Y neuroblastoma cells expressing mutant *parkin* showed a significant increase in the Ca^2+^ level, caused in particular, by Ca^2+^ release from ER [15], and a protective role of parkin against Ca^2+^ cytotoxicity in Drosophila [65] has been reported. Furthermore, the lack of Parkin or PINK1, which operates upstream of Parkin in the same pathway, results in impaired ubiquitination of Mfn [66], which plays an indispensable role in the formation of ER–mitochondria contact sites [67,68], causing a decreased tether between the ER and mitochondria. The high level of both cytosolic and mitochondrial Ca^2+^ in the Pt fibroblasts, as compared to the CTRL cells, might be a consequence of the increased activation of the RyR and IP3 receptors by PLCγ1, which are more active in the absence of parkin [15], or it might be correlated to the high basal levels of cAMP that are able to activate PKA and EPAC, which independently lead to an increase in the cytosolic concentration of Ca^2+^, modulating the release of Ca^2+^ from the ER by directly regulating the ryanodine receptors. Proteomics studies conducted on CRTL and *parkin*-mutant fibroblasts have shown the involvement of parkin in the regulation of Ca^2+^-binding proteins, which were downregulated in Pt fibroblasts [19].

To assess the contribution of ER to the cellular Ca^2+^ homeostasis, in the Pt fibroblast context, we tested the effects of thapsigargin, a specific irreversible inhibitor of all SERCA isoforms and dantrolene, which inhibits the RyR receptor. In both cases, following the treatment, major differences between CTRL and Pt cell Ca^2+^ levels were found in the mitochondria. Notably the changes observed in the cytosolic Ca^2+^ of treated Pt fibroblasts were not reflected in the mitochondrial Ca^2+^, despite them sharing the same responses to the MCU inhibitor observed in the CTRL fibroblasts. Physical contacts between the ER and mitochondria, called mitochondria-associated membranes (MAMs), provide a controlled Ca^2+^ flux from the ER store into the mitochondria without raising their concentration in the cytosol [69,70,71]. The loss or alteration of these ER–mitochondria contact sites have been associated with several neurodegenerative diseases, including Parkinson’s disease [72,73,74,75,76]. Recent studies showed altered MAMs in parkin-silenced SH-SY5Y cells [16], and endoplasmic reticulum–mitochondria interface perturbation in fibroblasts from Parkin (PARK2) knock-out mice, and from PD patients harboring PARK2 mutations [14]. The altered interactions between RE and the mitochondria, therefore, could be responsible for impaired mitochondrial Ca^2+^ homeostasis negatively affecting mitochondrial functions and biogenesis.

Taking into account the higher steady-state Ca^2+^ and cAMP levels in Pt fibroblasts, we investigated whether the two events were causally linked. To this aim, we reduced the mitochondrial Ca^2+^ level by treating both the Pt and CTRL cells with RR, and we found a specific reduction of cAMP only in the Pt fibroblasts. Further we mimicked an increased mitochondrial Ca^2+^ content, exploiting the effect of dantrolene treatment in the CTRL fibroblasts. Interestingly, in these conditions, we detected an increase of cAMP level, with a major contribution by sAC, and furthermore, a higher ADCY10, as well as PDE4A mRNA expression level, to balance the cAMP increase. This feedback regulation of cAMP levels, expected in CTRL cells, might be absent or dysregulated in Pt fibroblasts.

## 5. Conclusions

In conclusion, we showed for the first time to our knowledge, a significantly high basal level of cAMP in *parkin*-mutant fibroblasts, resulting from a greater contribution by sAC, and/or a low expression of PDE4, whose loss of function could be, however, compensated by other members of the PDE family, as shown by the results of IBMX treatment. Furthermore, our study confirms, as previously described, the involvement of parkin in the control of cellular Ca^2^+ levels, suggesting altered interactions between the endoplasmic reticulum and the mitochondria, and a potential different efficiency for the main Ca^2+^ transport mechanisms. An additional element to the link between parkin and mitochondria-related calcium homeostasis is provided by a recent study, showing that parkin selectively regulates the turnover of the MCU accessory protein MICU1 [77]. MICU1 is a regulatory subunit of the MCU complex that is found to exert a stimulatory effect on mitochondrial Ca^2+^ uptake [78]. Parkin promotes the proteasome-mediated degradation of MICU1, by direct interaction with the protein, and by a mechanism that is partially independent of its ubiquitination [77]. On this basis, it is conceivable that the loss of function in a parkin-null context enhances the MCU-mediated entry of Ca^2+^ into the mitochondria. The mitochondrial calcium overload could lead to the activation of sAC and PKA, via ensuing the elevation of the steady-state cAMP content as a possible compensatory mechanism to counteract Ca^2+^-induced ROS production, and the accumulation of damaged mitochondria (Figure 9).

PKA was found to induce phosphorylation-mediated activation of the respiratory chain complexes I and IV, thus enhancing mitochondrial respiration [79,80]. Though this mechanism is beneficial in non-pathological conditions, it may result in an unnecessarily permanent fostering of oxidative metabolism, with a slight but continuous overproduction of reactive oxygen species. This would lead to oxidative modifications of the respiratory chain components, which in turn would generate further ROS, thus establishing a pro-oxidative loop [10,11] in line with the reported evidences of a causal link between the alteration of cellular redox homeostasis, and the development of Parkinson’s disease.

## Figures and Tables

**Figure 1 cells-08-00250-f001:**
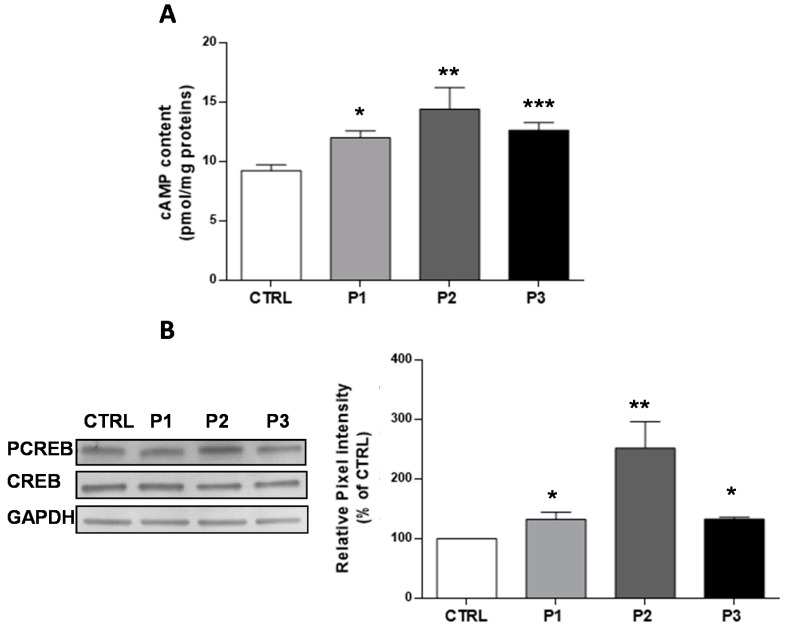
Basal cyclic adenosine monophosphate (cAMP) cellular levels and CREB phosphorylation status in total cell lysates. (**A**) Basal cAMP levels expressed as pmol/mg protein, in CTRL (open bar), P1 (light grey bar), P2 (dark grey bar), and P3 (black bar) fibroblasts. Data are means ± SEM from eight to 20 independent experiments; significance was calculated with a one-way ANOVA with a Bonferroni post hoc test; * *p* ˂ 0.05, ** *p* < 0.005, *** *p* ˂ 0.0001 vs. CTRL cells; (**B**) Representative Western blot of CREB and PCREB, performed on whole cell lysates. The bar graph shows the PCREB/CREB ratio, calculated by the quantification by the densitometric analysis of band intensity normalized to GAPDH, and used as the loading control. Data, expressed as a percentage of CTRL values, are means ± SEM from 3 to 6 independent experiments; significance was calculated with one-way ANOVA with Bonferroni post hoc tes; ** *p* < 0.005 vs. CTRL cells.

**Figure 2 cells-08-00250-f002:**
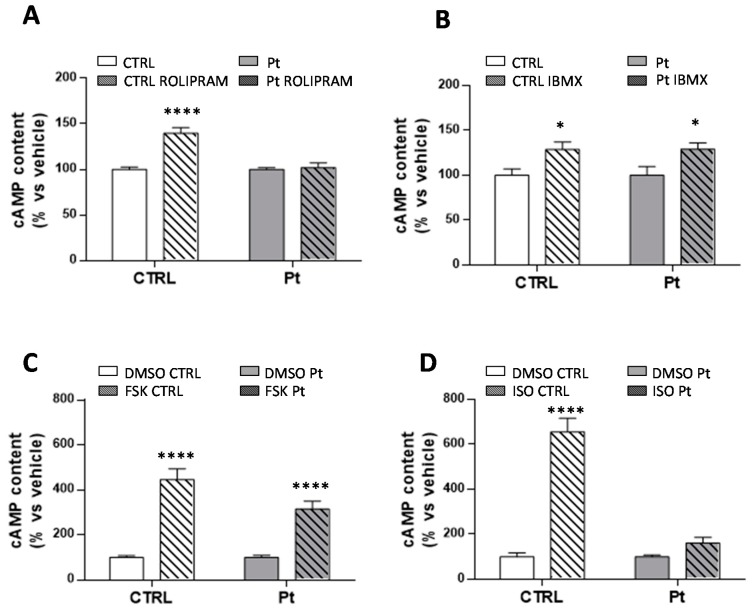
cAMP cellular levels affected by the regulation of PDE and AC activity. (**A**–**D**) cAMP content in CTRL (open bar) and Pt (gray bar) fibroblasts incubated with ROLIPRAM (**A**), IBMX (**B**), FSK (**C**), and ISO (**D**). Data, expressed as a percentage of DMSO values, are means ± SEM from four to six independent experiments under each condition. Significance was calculated with one-way ANOVA with Bonferroni post hoc test; * *p* ˂ 0.05, **** *p* ˂ 0.0001 vs. vehicle-treated cells. See Materials and Methods for experimental details.

**Figure 3 cells-08-00250-f003:**
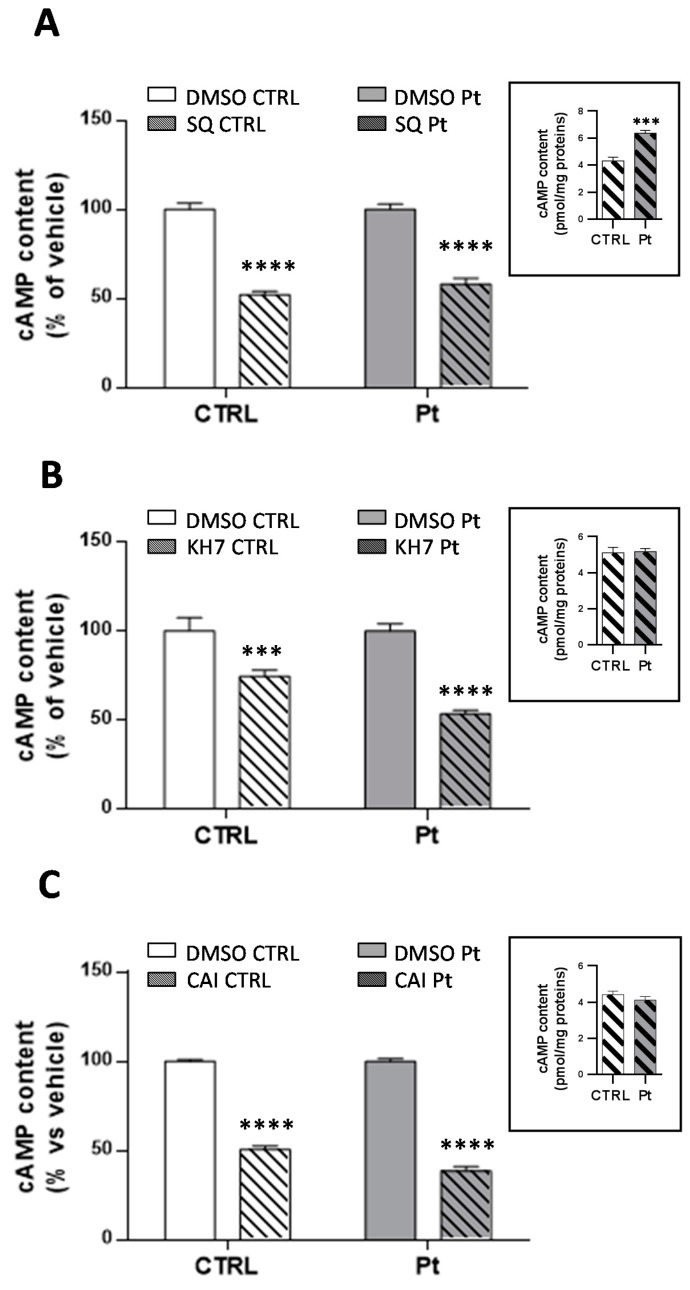
The contribution of tmAC and sAC to the cAMP cellular levels. cAMP content in the CTRL (open bar) and Pt (grey bar) fibroblasts incubated with SQ22386 (**A**), KH7 (**B**) and CAI (**C**). Data, expressed as a percentage of the DMSO values, are means ± SEM from three to 14 independent experiments under each condition. The significance was calculated with one-way ANOVA with a Bonferroni post hoc test; **** *p* ˂ 0.0001 vs. vehicle-treated cells. See Materials and Methods for experimental details. The bar graphs in the inset of (**A**–**C**) display the absolute values of the cAMP content after drug treatment, expressed as pmol/mg protein. Values are means ± SEM from three to 14 independent experiments; the significance was calculated with the Student’s *t* test. *** *p* ˂ 0.0005, **** *p* ˂ 0.0001 vs. CTRL-treated cells.

**Figure 4 cells-08-00250-f004:**
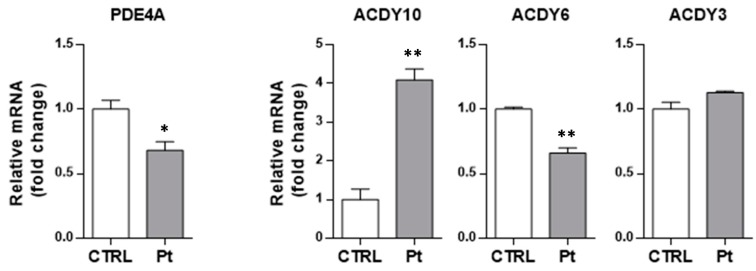
Transcription levels of PDE4A, sAC, and tmAC isoforms in fibroblasts. mRNA levels of PDE4, ADCY10 (sAC), ADCY6 (tmAC), and ADCY3 (tmAC), normalized to the housekeeping gene GAPDH, by qRT-PCR analysis (ΔΔCT) in CTRL (open bars) and Pt (grey bars) cells. Data, expressed as fold-changes mRNA expression levels in Pt fibroblasts, compared to CTRL cells, are mean ± SEM of three independent experiments performed in duplicate. See Materials and Methods for experimental details. Significance was calculated with the Student’s *t*-test; * *p* < 0.05, ** *p* < 0.005 vs. CTRL cells.

**Figure 5 cells-08-00250-f005:**
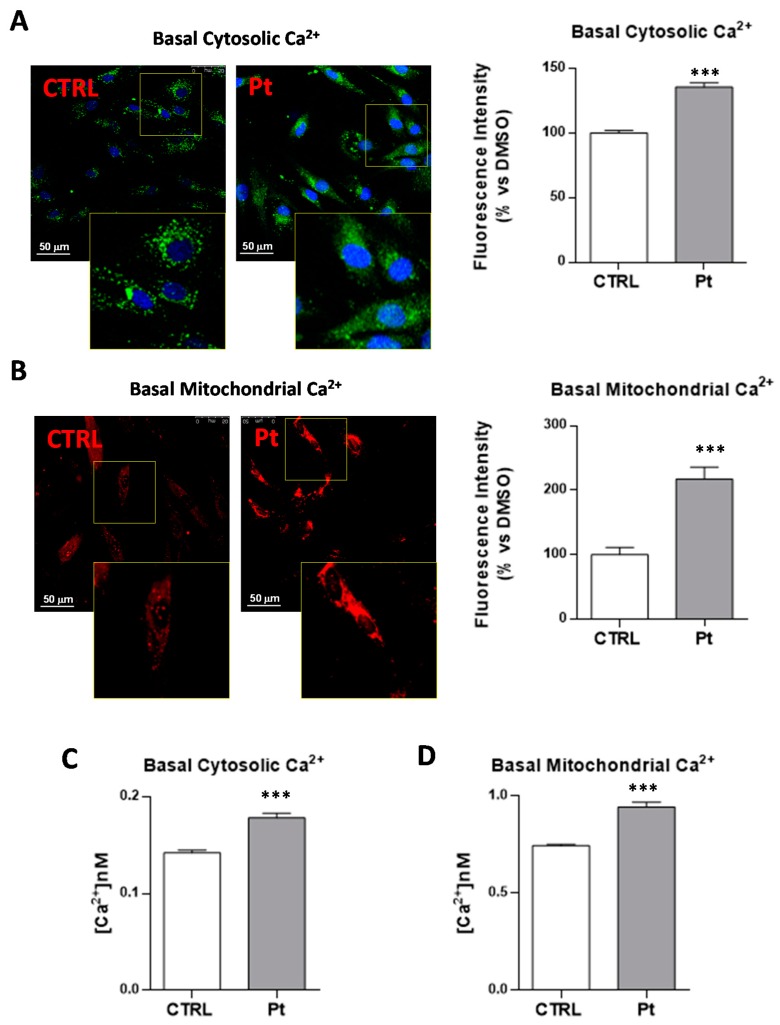
Cytosolic and mitochondrial Ca^2+^ levels in fibroblasts cultures. CTRL and Pt fibroblasts cultures were incubated for 30 min with 2 µM Fluo-4AM for cytosolic Ca^2+^ or 4 µM X-Rhod-1AM for mitochondrial Ca^2+^, and cell nuclei were stained with 1 μg/mL DAPI. The analysis was performed by confocal microscopy, as described in Materials and Methods. *LSCM* representative images of basal cytosolic (**A**) and mitochondrial (**B**) Ca^2+^ levels monitored in CTRL and Pt live cells, representative of five different preparations for each condition. Scale bars, 50 µm. The graph displays a quantitative analysis of the probe-related fluorescent signal mean intensities, with the bars indicating values that are averaged from four to seven different optical fields of three independent experiments under each condition (±SEM). Data are expressed as a percentage of vehicle-treated cells; significance was calculated with Student’s *t* test; *** *p* ˂ 0.0001 vs. vehicle-treated cells. See Materials and Methods for the experimental details. Spectrofluorimetric measurements of cytosolic (**C**) and mitochondrial (**D**) Ca^2+^ contents in CTRL (open bars) and Pt (grey bar) cells loaded, respectively, with Fluo-4AM and X-Rhod-1AM. Data, means ± SEM from seven to 12 independent experiments under each condition, are expressed as [Ca^2+^] nM. Significance was calculated with Student’s *t*-test; *** *p* ˂ 0.0001 vs. vehicle-treated cells. See Materials and Methods for the experimental details.

**Figure 6 cells-08-00250-f006:**
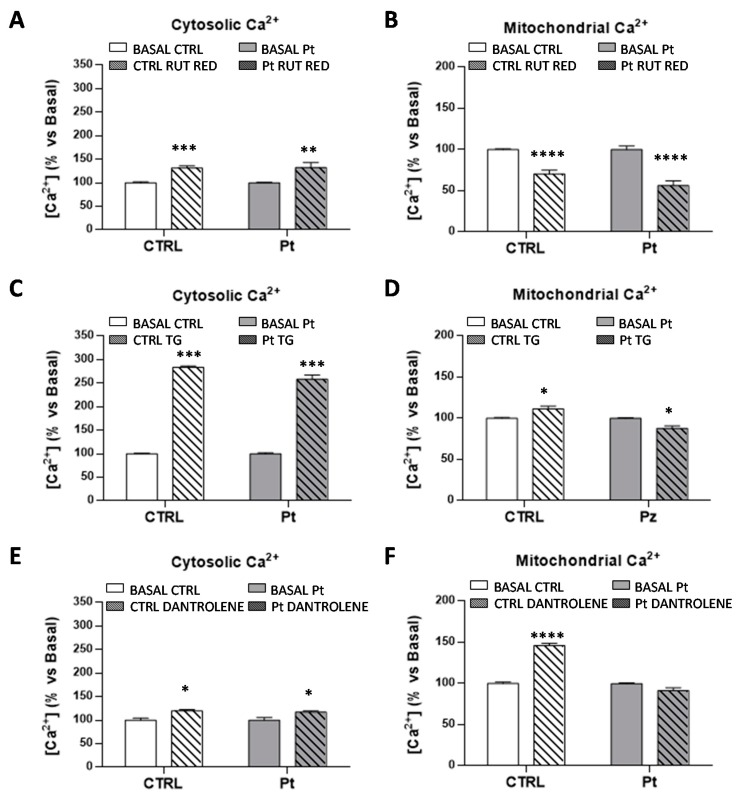
Mitochondrial and endoplasmic contribution to the basal cellular Ca^2+^ levels. Cytosolic and mitochondrial Ca^2+^-spectrofluorimetric measurements in CTRL (open bar) and Pt (grey bar) cells, monitored respectively by cytosolic Ca^2+^ probe Fluo-4AM (**A**,**C**,**E**) and mitochondrion Ca^2+^ probe X-Rhod-1 AM (**B**,**D**,**F**). The graphs display the Ca^2+^ content in CTRL (open bar) and Pt (grey bar) fibroblasts after pre-treatment with Ca^2+^ flux inhibitors: (**A**,**B**) Ruthenium Red (RR), (**C**,**D**) Thapsigargin (TG), and (**E**,**F**) Dantrolene. Data, means ± SEM of three independent experiments, are expressed as a percentage of the vehicle-treated cells. See Materials and Methods for the experimental details. Significance was calculated with one-way ANOVA, with Bonferroni post hoc test; * *p* ˂ 0.05, ** *p* ˂ 0.005, *** *p* ˂ 0.0005, **** *p* ˂ 0.0001 vs. vehicle-treated cells.

**Figure 7 cells-08-00250-f007:**
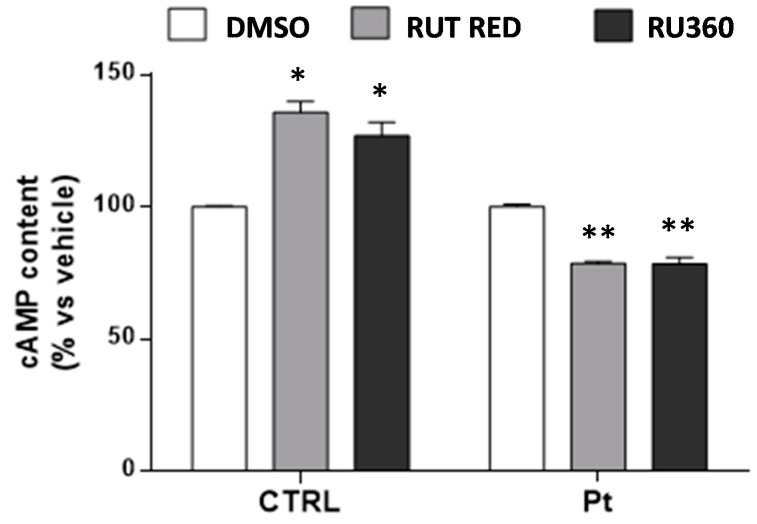
Mitochondrial Ca^2+^ contribution to the cellular cAMP content. cAMP content in the CTRL and Pt fibroblasts incubated with RR and Ru360. Data, means ± SEM from three independent experiments under each condition, are expressed as percentage of vehicle-treated cells. The significance was determined by one-way ANOVA analysis with Bonferroni post hoc test; * *p* ˂ 0.05, ** *p* ˂ 0.005 vs. vehicle-treated cells. See the Materials and Methods for experimental details.

**Figure 8 cells-08-00250-f008:**
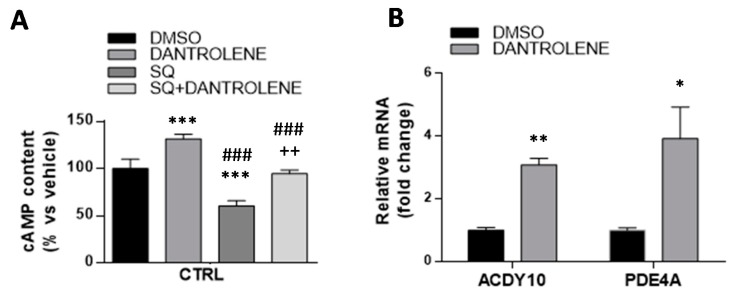
cAMP levels in dantrolene-treated CTRL cells. (**A**) cAMP content in the CTRL fibroblasts incubated with 10 µM Dantrolene or 100 µM SQ22536 and co-incubated with Dantrolene + SQ22536 for 30 min at 37 °C. Data, means ± SEM of three independent experiments, are expressed as a percentage vs. vehicle-treated cells (black bar). See Materials and Methods for the experimental details. The significance was determined by one-way ANOVA analysis with Bonferroni post hoc test; *** *p* < 0.0001 vs. vehicle-treated cells, ### *p* < 0.0001 vs. dantrolene-treated cells, ++ *p* < 0.005 vs. SQ22536-treated cells. (**B**) Semi-qRT-PCR analysis of PDE4A and ADCY10 (sAC) mRNA levels normalized to the housekeeping gene GAPDH (ΔΔCT), in vehicle-treated (black bar), and dantrolene-treated (grey bar) CTRL cells. Data, expressed as fold change mRNA expression levels in dantrolene-treated compared to vehicle-treated CTRL cells, are mean ± SEM of three independent experiments performed in duplicate. See Materials and Methods for the experimental details. Significance was calculated with Student’s *t*-test; * *p* < 0.05, ** *p* < 0.005 vs. vehicle-treated cells.

**Figure 9 cells-08-00250-f009:**
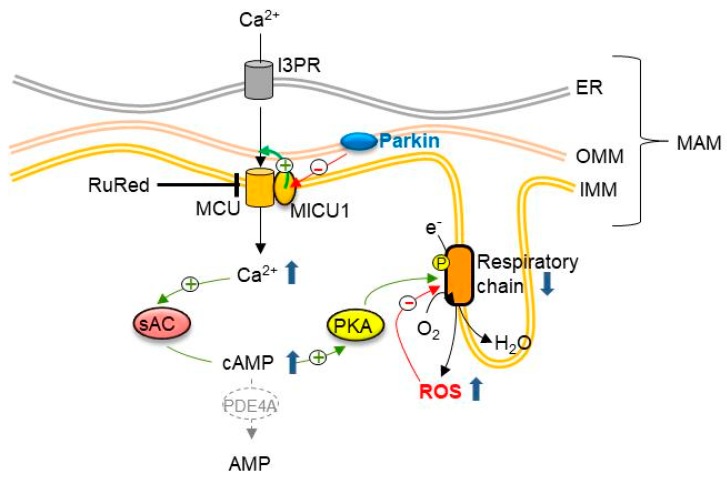
Drawing of the deregulation of sAC-dependent cAMP and Ca^2+^ homeostasis in *parkin*-mutant fibroblasts. The scheme summarizes the main findings that are reported in this study, and suggests the link between parkin and the homeostasis of Ca^2+^ at mitochondria-associated membranes (MAMs), mitochondrial cAMP, and redox balance. See the Discussion for details.

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
