# Peer review of "Increased Levels of cAMP by the Calcium-Dependent Activation of Soluble Adenylyl Cyclase in Parkin-Mutant Fibroblasts"

_cells, 2019, doi:10.3390/cells8030250_

Round 1
Reviewer 1 Report
The manuscript by Tanzarella et. al. describes the dissection of calcium and cAMP pathways in Parkin-mutant (Pt) fibroblasts isolated from patients with early onset Parkinson's disease. The authors observe elevated basal cAMP levels in Pt cells as compared to control cells along with corresponding differences in pCREB levels. They are able to explain this finding by increased expression of sAC and decreased expression of PDE4A, along with increased calcium in the cytosol and mitochondria. These are novel findings, however no direct link between Parkin and the altered calcium signaling is demonstrated.
Specific comments:
1. While this manuscript makes some interesting advances, it does not demonstrate a causal link between Parkin and the increased calcium/sAC/PDE4 alterations. Parkin mutation causes significant mitochondrial abnormalities (as stated in the Introduction) so there are likely many factors at play that influence calcium and cAMP signaling. The authors need to be very careful in drawing conclusions and should add discussion about the many other possible linkages.
2. The cAMP methods used are not clear. In one place it is stated that a kit from Enzo was used and later it says cAMP was detected with a kit from Assay Designs). Also, the data is normalized to total protein content but there is no description of how protein was measured. This key methodology needs to be clarified.
3. Page 5, first paragraph contains many incorrect references. Line 160 has a reference to tmAC but reference 30 is a review of sAC. This needs to be changed to a recent review of tmAC's (the most recent one is Pharmacol Rev. 2017 Apr;69(2):93-139). Several of the other references in this section also appear to be incorrect.
4. Figure 1B shows densitometry of pCREB/CREB ratio but a representative immunoblot image also needs to be shown.
5. Figure 3C does not show any statistical analysis of the comparison that is most important: the difference between CTRL and Pt. Without this there is now way to know if acetazolamide had a different effect in the Pt cells.
6. It is unclear why the author focused on PDE4A in their analysis of PDE expression. There are many PDE genes and alteration in any of the cAMP-hydrolyzing isoforms could contribute to the observed effects. Data on the expression of the other PDE isoforms needs to be shown.
7. Line 249, reference to Figure 7A is incorrect.
8. Numerous grammatical and syntax errors exist, so a review for proper English is needed.
Author Response
Reviewer #1
Specific comments:
1. While this manuscript makes some interesting advances, it does not demonstrate a causal link between Parkin and the increased calcium/sAC/PDE4 alterations. Parkin mutation causes significant mitochondrial abnormalities (as stated in the Introduction) so there are likely many factors at play that influence calcium and cAMP signaling. The authors need to be very careful in drawing conclusions and should add discussion about the many other possible linkages.
Following the reviewer’s suggestion, we have taken into account, in the discussion section, some further possible linkage mechanisms between parkin loss of function and the increased calcium and sAC related cAMP level.
[page 18-19, lines 433-442].
2. The cAMP methods used are not clear. In one place it is stated that a kit from Enzo was used and later it says cAMP was detected with a kit from Assay Designs). Also, the data is normalized to total protein content but there is no description of how protein was measured. This key methodology needs to be clarified.
We apologize for the oversight. Only one cAMP assay kit was used throughout the study. We have, therefore, modified that part of the Methods and, in addition, indicated the protocol used to measure the protein concentration detection in a new paragraph of Materials and Methods
[page 3, lines 84-95 and page 4 lines 149-151]
3. Page 5, first paragraph contains many incorrect references. Line 160 has a reference to tmAC but reference 30 is a review of sAC. This needs to be changed to a recent review of tmAC's (the most recent one is Pharmacol Rev. 2017 Apr;69(2):93-139). Several of the other references in this section also appear to be incorrect.
We apologize for the misappropriation and we have changed the reference 30 as requested (number 29 in the revised version). We took also care as well of others references.
[page 5, lines 184-186]
4. Figure 1B shows densitometry of pCREB/CREB ratio but a representative immunoblot image also needs to be shown.
Following the reviewer’s suggestion, Figure 1B has been updated with immunoblot images.
[page 8]
5. Figure 3C does not show any statistical analysis of the comparison that is most important: the difference between CTRL and Pt. Without this there is now way to know if acetazolamide had a different effect in the Pt cells.
Following the reviewer’s suggestion, Fig 3C has been updated with the inset showing comparison between CAI- treated CTRL and Pt cells.
[page 11]
6. It is unclear why the author focused on PDE4A in their analysis of PDE expression. There are many PDE genes and alteration in any of the cAMP-hydrolyzing isoforms could contribute to the observed effects. Data on the expression of the other PDE isoforms needs to be shown.
We agree with the reviewer. There are many PDE isoforms but the PDE4 isoform is the major cAMP degrading enzyme in fibroblasts [Selige J, Hatzelmann A, Dunkern T. The differential impact of PDE4 subtypes in human lung fibroblasts on cytokine-induced proliferation and myofibroblast conversion. J Cell Physiol. 2011 Aug;226(8):1970-80; Schafer PH, Truzzi F, Parton A, Wu L, Kosek J, Zhang LH, Horan G, Saltari A, Quadri M, Lotti R, Marconi A, Pincelli C. Phosphodiesterase 4 in inflammatory diseases: Effects of apremilast in psoriatic blood and in dermal myofibroblasts through the PDE4/CD271 complex. Cell Signal. 2016 Jul;28(7):753-63]. Furthermore, we got indirect information of the others cAMP-hydrolyzing isoforms activity from the IBMX treatment experiments which sustained the evidence that similar PDEs activity is present in both CTRL and Pt cells. A comment has been added to the text.
[page 18, lines 411-422]
7. Line 249, reference to Figure 7A is incorrect.
We apologize for the misappropriation and we have modified the reference to Figure 8A.
[page 7, line 276]
8. Numerous grammatical and syntax errors exist, so a review for proper English is needed.
We apologize for the syntax oversights and grammar errors. The paper has been carefully revised to improve the grammar and readability.

Reviewer 2 Report
see attached file (critique.pdf).

Author Response
Reviewer #2
Specific comments:
1. L73. Informed consent was mentioned. Is there IRB approval?
The study was approved by the local Ethics Committee at the University of Bari Medical School and the sentence has been reported in Materials and Methods.
[page 2, lines 77-78]
2. L90-98. The heading says “live cell imaging” but cells were fixed. Need clarification. In addition, Fig. 5 pictures show nuclear staining in blue, which was not described.
Accordingly, we modified the heading specifying that the analysis was performed on fixed cells and added the nuclei staining with DAPI both in Materials and Methods section 2.3 [page 3, lines 95-105] and in the legend to Figure 5 [page 13, line 337]
3. L113-117. Need more detail. The cited reference does not have any detail. GAPDH primers are not presented. b-actin was not used in the manuscript.
According to the proper observation of the reviewer we have modified the description of Real Time PCR in Material and Methods as follows: Purification of total RNA from fibroblasts was carried out using RNeasy Mini Kit (Qiagen), according to the manufacturer's protocol. One microgram of total RNA was then reverse-transcribed to generate cDNA for PCR by using iScript cDNA Synthesis kit (Bio-Rad). Semi quantitative determination of ADCY and PDE mRNA levels was performed by real-time qRT-PCR using SYBR Green (Bio-Rad). Reactions were performed in duplicate for each sample in three independent experiments. The multiple reactions were performed in a volume of 25 μl containing 12.5 μl of 2×PCR master mix, 0.2 μM of specific primers and 200 ng of cDNA template. Amplifications were performed in BioRad iCycler iQ Real-Time PCR Detection System using the following cycle program: denaturation step at 95°C for 10 min followed by 40 cycles of denaturation at 95°C for 15 s, annealing at 60°C for 1 min and extension at 72°C for 30 sec. The relative mRNA expression levels were calculated using the comparative CT method (ΔΔCT) described by Livak and Schmittgen [Methods. 2001 Dec;25(4):402-8. Analysis of relative gene expression data using real-time quantitative PCR and the 2(-Delta DeltaC(T)) Method. Livak KJ1, Schmittgen TD]. Quantitative normalization for each sample was performed using glyceraldehyde-3-phosphate dehydrogenase (GAPDH) as an internal control to normalize the variability in expression levels. Validated primers for semi - qRT-PCR are provided in Table S1.
[page 4, lines 122-137]
4. L141. “iper-phosphorylation” Typo?
As recommended, we corrected the typo in the text.
[page 5, line 165]
5. L145-147. Western blot should be presented.
Following the reviewer’s suggestion, Figure 1B has been updated with immunoblot images.
[page 8]
6. L155-157. IBMX treatment resulted in the similar level of cAMP increase in both CTRL and Pt cells, indicating that similar level of PDE activity is present in the both cell types regardless of PDE4. This is not noted in the later discussion.
We agree with the review and as suggested we have strengthened the evidence that similar PDEs activity is present in both CTRL and Pt fibroblasts except for the PDE4 isoform, whose mRNA expression level is reduced. A comment has been added to the text.
[page 18, lines 415-422]
7. L162-169. Forskolin effect on Pt cells was reduced about 40%. In addition, ISO had effect only on CTRL cells, suggesting alteration in tmAC. Other Gs-coupled ligands or cholera toxin should be used to examine further. To suggest alteration in adrenergic receptor signaling is premature.
As recommended, we modified the text avoiding to suggest alteration in adrenergic receptor signalling.
[page 5, lines 192-193 and page 18, lines 403-410]
8. L170-177. In general, basal activity of tmACs is very low. In order to examine the tmAC activity, they should be activated with Gs activation and/or forskolin.
Although we agree with the reviewer, the purpose of our study was to measure cAMP levels under resting conditions to assess whether there were differences between control and parkin-mutant fibroblasts.
9. L178-181. This reviewer expected that acetazolamide treatment increases the activity of sAC as published (Invest Ophthalmol Vis Sci. 2014;55:187-197), if cAMP measurement was done in the culture medium under 5% CO2. Need explanation.
The paper indicated by the reviewer refers to a study carried out with nonpigmented ciliary epithelial cells where a transient increase of the cAMP level was observed 2-5 min after the addition of the CAI inhibitors. The different result attained might be due to the different cell type tested and/or to the longer drug-treatment in our study (i.e. 30 min). However, the sAC-dependent cAMP decrease reported in our study is similar to what shown in other papers where the activity of sAC was inhibited by preventing the generation of HCO3− from CO2 with acetazolamide (CAI) in cells cultured under 5% CO2 [Strazzabosco M, et al. Differentially expressed adenylyl cyclase isoforms mediate secretory functions in cholangiocyte subpopulation. Hepatology. 2009; 50:244–252; Acin-Perez R, et al. Cyclic AMP Produced inside Mitochondria Regulates Oxidative Phosphorylation. Cell Metabolism. 2009; 9:265–276] or in isolated mitochondria [De Rasmo D, et al. Intramitochondrial adenylyl cyclase controls the turnover of nuclear-encoded subunits and activity of mammalian complex I of the respiratory chain. Biochim Biophys Acta. 2015 Jan;1853(1):183-9].
10. L183-194. It is not clear why the authors selected two tmAC isoforms out of nine. What is the y-axis unit? According to the legend it must be relative to GAPDH. It is very odd to see the expression of all four genes are very similar to that of GAPDH, whose expression is pretty high. Need explanation.
We performed analysis of mRNA expression of ADCY3 and ADCY6, since are two tmAC isoforms widely expressed in various human tissues and belong to different distinct families based on their amino acid sequence similarity, functional attributes and regulation. ADCY3, is a protein involved in a number of physiological and pathophysiological metabolic processes and ADCY6 an isoform that belongs to a small subclass of adenylyl cyclase proteins functionally related [Defer N, Best-Belpomme M, Hanoune J. Tissue specificity and physiological relevance of various isoforms of adenylyl cyclase. Am J Physiol Renal Physiol. 2000 Sep;279(3): F400-16; Liu X, Li F, Sun SQ, Thangavel M, Kaminsky J, Balazs L, Ostrom RS. Fibroblast-specific expression of AC6 enhances beta-adrenergic and prostacyclin signaling and blunts bleomycin-induced pulmonary fibrosis. Am J Physiol Lung Cell Mol Physiol. 2010 Jun;298(6): L819-29].
Thanks for the suggestion for a better interpretation what presented in Figures 4 and 8B. The y-axis shows the mRNA levels as fold changes relative to CTRL mRNA levels. We have modified the y-axis label in Figures 4 and 8B.
[page 13 and 16]
11. L212. The text does not match with the figure.
We apologize for the oversight and have modified the text as recommended.
[page 6, lines 237-239].
12. L213. Active uptake is misleading. MCU does not perform active transport.
We wanted to indicate ‘… an efficient mitochondrial Ca2+ uptake by this uniporter …’. To avoid any misinterpretation we replaced ‘active’ with ‘efficient’.
[page 6, line 239].
13. L259-263. According to the legend and M & M, the duration of the treatment was 30 min. Do the authors claim that the mRNA expression level increase 3-4 fold in 30 min and the newly synthesized proteins (sAC and PDE4) affect the cAMP level?
We thank the reviewer for the comment. We didn’t want to claim that the increase of mRNA expression level affected the cAMP level. For this reason and to better explain our point of view we changed the sentence as follows:
In order to assess whether the high calcium level measured in dantrolene-treated CTRL might led to a differential ADCY10 and PDE4A gene expression, as observed in Pt cells, we measured mRNA levels. In this setting, the ADCY10 mRNA levels resulted significantly higher (Figure 8B), as well as the expression of PDE4A, maybe to counteract the observed cAMP increase, as you would expect in healthy cells.
[page 7, lines 286-292].
14. L276. Typo?
As recommended, we corrected the typo in the text.
[page 8, line 309]
15. Figure 4. Does the level 1.0 in Y-axis correspond to the same amount as GAPDH?
As described in point 10 the level 1.0 in Y-axis doesn’t correspond to the same amount as GAPDH but the y-axis shows the mRNA levels as fold changes relative to CTRL mRNA levels. We have modified the y-axis label in Figures 4 and 8B.
[page 13 and 16]
16. L365-368. This argument is very weak and not supported by forskolin data. There was a significant reduction of forskolin-activated cAMP in Pt cells. Other Gs-coupled ligands and/or cholera toxin should have used to confirm this argument.
As reported in point 7 we agree with the argument raised by the reviewer that alteration in adrenergic receptor signalling is premature and, consequently, have modified the text avoiding to suggest alteration in adrenergic receptor signalling.
[page 5, lines 192-193 and page 18, lines 403-410]
17. L388. “The high basal levels of mitochondrial cAMP in Pt fibroblasts”. This text is not correct. The authors never showed intracellular location of cAMP or sAC.
As recommended, we modified the text.
[page 18, line 433]
18. L401. “in the Pt fibroblast contest”. Typo?
As recommended, we corrected the typo in the text.
[page 7, line 275]
19. L418-419. See comment #17.
As recommended, we modified the text.
[page 19, line 468]
20. L425-428. “a low expression of PDE4, whose loss of function cannot be compensated by other members of PDE families”. This does not match the results presented in Figure 2B.
We agree with the review and we have modified the text.
[page 19, lines 477-479]
21. L436. “in a parkin-null contest”. Typo?
As recommended, we corrected the typo in the text.
[page 20, lines 488]

Round 2
Reviewer 1 Report
The authors have significantly improved their manuscript and addressed reviewer concerns. However, some minor English editing is still required in some parts.
Author Response
The text has been revised by professional English Language Expert
Reviewer 2 Report
The authors addressed the concerns raised by reviewers. However, there are still some issues remain.
LSCM imaging. The fluorescent dyes for calcium measurement are commonly used for live cell imaging. I wonder if cytosolic and mitochondrial calcium remains unchanged after fixing and washing. It is odd considering all staining and imaging can be performed using live cells.
Statistical analyses. Data presented in Figure 1 should be analyzed by one way-ANOVA. As for data presented in Figures 2, 3, and 6, I am not sure if two-way ANOVA is the appropriate analysis since data were normalized to control values.
Semi-qRT-PCR analysis. Data presented in Figure 4 should be relative amounts to GAPDH so that the differences among the levels of four mRNA expression can be observable.
L132. Need reference.
L215. Is this 4 fold higher?
Author Response
1. LSCM imaging. The fluorescent dyes for calcium measurement are commonly used for live cell imaging. I wonder if cytosolic and mitochondrial calcium remains unchanged after fixing and washing. It is odd considering all staining and imaging can be performed using live cells.
We thank the reviewer for this pertinent observations, however we are confident in our results because these semiquantitative analysis are confirmed by quantitative spectrofluorimetric measurements performed in living cells.
2.Statistical analyses. Data presented in Figure 1 should be analyzed by one way-ANOVA. As for data presented in Figures 2, 3, and 6, I am not sure if two-way ANOVA is the appropriate analysis since data were normalized to control values.
As you suggested, we performed one way-ANOVA statistical test.
3. Semi-qRT-PCR analysis. Data presented in Figure 4 should be relative amounts to GAPDH so that the differences among the levels of four mRNA expression can be observable.
Following you can find all the details about our analysis showing that the mRNA level of the gene of our interest is normalized to mRNA level of GAPDH.
RELATIVE quantification of gene expression
The cycle thresholds (ct) of the genes of interest were compared to cycle thresholds of the housekeeping genes (GAPDH) to determine relative changes in gene expression.
For example
1. Normalize CT of target gene to CT reference gene: ∆CT
Sample | Gene | ∆Ct | |
CT target (PDE4 ) | CT reference (GAPDH) | CT Target- CT Reference | |
control (Calibrator) | 30,05 | 20,15 | 9,9 |
Pt (test) | 30,19 | 19,47 | 10,72 |
2. Normalize ∆CT of the test sample to ∆CT of calibrator: ∆∆CT
∆∆CT = ∆CT (test)- ∆CT (Calibrator)= 10.72 - 9.9 = 0.82
3. Calculate fold change in expression:
2-∆∆CT = normalized expression ratio 2- 0.82= 0.56
Pt cells express PDE4 lower than control cells
4. L132. Need reference.
The reference has been added.
5. L215. Is this 4 fold higher?
The sentence has been revised.